# The Molecular Bases Study of the Inherited Diseases for the Health Maintenance of the Beef Cattle

**DOI:** 10.3390/genes12050678

**Published:** 2021-04-30

**Authors:** Elena Konovalova, Olga Romanenkova, Olga Kostyunina, Elena Gladyr

**Affiliations:** L.K. Ernst Federal Research Center for Animal Husbandry, 142132 Moscow Region, Russia; ksilosa@gmail.com (O.R.); kostolan@yandex.ru (O.K.); elena-gladyr@mail.ru (E.G.)

**Keywords:** genetic defects, congenital diseases, meat cattle, gene mutations, Aberdeen Angus, Hereford, Simmental, Belgian Blue

## Abstract

The article highlighted the problem of meat cattle genetic defects. The aim was the development of DNA tests for some genetic defects diagnostics, the determination of the animal carriers and their frequencies tracking in time. The 1490 DNA samples from the Aberdeen Angus (*n* = 701), Hereford (*n* = 385), Simmental (*n* = 286) and Belgian Blue (*n* = 118) cattle have been genotyped on the genetic defects by newly created and earlier developed DNA tests based on AS-PCR and PCR-RFLP methods. The Aberdeen Angus cattle genotyping has revealed 2.38 ± 0.31% AMC-cows and 1.67 ± 0.19 % AMC-bulls, 0.65 ± 0.07% DDC-cows and 0.90 ± 0.10% DDC-bulls. The single animals among the Hereford cattle were carriers of MSUD and CWH (on 0.27 ± 0.05%), ICM and HY (on 0.16 ± 0.03%). The Simmental cattle were free from OS. All Belgian Blue livestock were M1- and 0.84%-CMD1-carriers. The different ages Aberdeen Angus cattle genotyping has shown the tendency of the AMC- and DDC frequencies to increase in the later generations. The statistically significant increase of DDC of 1.17% in the cows’ population born in 2019 compared to those born in 2015 allows concluding the further development of the DNA analysis-based measures preventing the manifestation of the genetic anomalies in meat cattle herds is necessary.

## 1. Introduction

The issue of providing people with food and food safety is relevant regardless of the time and geographical location of any state. One of the most important strategic objectives is to increase the consumption of meat products, in particular beef, as the main source of essential amino acids. In this regard, the efforts of specialists of the beef cattle industry are constantly directed at finding ways to increase the productivity of animals, and the use for this purpose of breeding material of specialized meat cattle breeds is very promising [1]. However, we must not forget that the manifestation of the properties of high productivity associated with the intense functioning of all organs and systems of the body that can be a prerequisite for the accumulation of genetic cargo in the form of mutant alleles associated with the manifestation of hereditary diseases [2]. The manifestation of congenital genetic pathologies can cause quite serious economic damage to farms, often in the form of stillborn calves, which reduces the yield of calves per cow, the main indicator of the profitability of beef cattle breeding.

Modern molecular biology over the world characters by the rapid development of genomic selection, allowing the selection of a decision based on genomic breeding values (GEBV) estimation with simultaneous parentage verification, imputation and genetic defects identification [3]. For the obtaining of genome information, the genotyping with the medium-density Bovine SNP50 v3 BeadChip (now standard in cattle) is available with more than 50,000 informative SNPs, or the high-density BovineHD Genotyping BeadChip (www.illumina.com (accessed date 11 March 2021)), which contains 777,609 single nucleotide polymorphisms (SNPs), have been used in the current time [4].

However, the use of the DNA chips is limited by the character of studied mutations and is appropriate for only SNP genotyping (but not for deletions, insertions or duplications). In some cases, the farmer is faced with the need to identify animals exclusively that carry genetic defects, and the use of DNA chips for only this purpose is not cost-effective.

Due to this fact, the use of genetic defects DNA-diagnostics based on the different polymerase chain reaction (PCR) modifications in addition to the genome sequencing methods is not losing actuality.

Disposing the knowledge about some gene mutations associated with the appearance of genetic diseases (https://omia.org/home/ (accessed date 11 March 2021), https://www.ncbi.nlm.nih.gov/ (accessed date 11 March 2021)), we have studied five genetic defects registered in Aberdeen Angus cattle (AM, OS, DD, MA and M1), five defects of Hereford cattle (OS, MSUD, ICM, HY and CWH), one defect of Simmental cattle (OS) and two defects of Belgian Blue cattle (M1 and CMD1). The short characteristics of the defects and caused mutations are presented in Table 1.

The aim of our work was the development of DNA tests for the diagnostics of some genetic defects appearing in Aberdeen Angus, Hereford, Simmental and Belgian Blue cattle breeds, the evaluation of the genotype frequencies of the animal carriers of mutant alleles and the tracking the percent of animal carriers in one population in time.

## 2. Materials and Methods

The DNA probes (*n* = 1490) from Aberdeen Angus (*n* = 701), Hereford (*n* = 385), Simmental (*n* = 286) and Belgian Blue (*n* = 118) cattle deposited in DNA collections of the L.K. Ernst Federal Research Center for Animal Husbandry were used as materials. The probes were obtained by sets of DNA extraction from the animal’s biomaterial (e.g., blood, skin, sperm), according to the manufacturer’s instructions (Extran 1- and 2, Syntol Co., Russia). The descriptions of the animals from which the DNA samples have been obtained are presented in Table 2. The cattle populations have belonged to the Central (the breeds of Simmental-1, 2, Belgian Blue and Aberdeen Angus-2a, 2b), Northern-West (the breeds of Aberdeen Angus (1a–1d) and Hereford (1a–1c)), Siberian (the breed of Simmental 3, 4) and the North-Caucasus (the breed of Hereford population 2) Federal Districts of Russia.

The Aberdeen Angus cattle DNA probes have been genotyped on AM, OS, DD and M1 genetic defects, the Hereford ones on OS, MSUD, ICM, HY and CWH genetic defects, the Simmental–on OS and the Belgian Blue–on M1 and CMD1 genetic defects.

The animal genotyping has been carried out using the AS-PCR (for the defects caused by deletion and duplication) and PCR-RFLP (for the defects caused by SNP) methods.

Genotyping has been carried out by using oligonucleotide primers as earlier designed (for AM, OS, DD, M1, MSUD, ICM and HY) [14,21] as newly created (for CWH and CMD1). Our oligonucleotide primers’ characteristics have been presented in Table 3.

The PCR was carried out on the thermocycler Biorad T100 (Bio-rad, Hercules, CA, USA) in volumes of 10–15 microliters under the next conditions: initial denaturation—3 min, 35 cycles of denaturation-annealing-elongation (45-30-40 s) and final elongation—4 min. The temperatures of denaturation and elongation were 95 °C and 72 °C, respectively; the annealing temperatures depended on the mutations (Table 3).

For SNP diagnostics, the PCR-RFLP method was chosen for CMD1 based on the further digestion of the PCR-fragment of the enzyme (restriction endonuclease). The choice of restriction sites was performed by the NEBcutter V2.0 Software (https://nc2.neb.com/NEBcutter2/ (accessed date 11 March 2021)). For the mutant allele associated with the defect, the restriction site for restriction endonuclease *DraIII* (restriction site CACNNN**↑**GTG) was found. The validation of PCR-RFLP for CMD1 and DD diagnostics earlier was conducted by Sanger sequencing (Eurogene Co., Moscow, Russia) with the assessment of the results by UGENE Software.

The amplification products were accessed by electrophoresis in agarose gel with content of 3% agarose under 120 V for 30 min.

The frequencies of the animal carriers’ genotypes was counted by formula (1):(1)$$f=\frac{N_{AC}}{N}\times 100$$
where *f* is genotype frequency; *N_AC_* is number of animal carriers; *N* is the total number of the population.

The average frequency was counted by formula (2):(2)$$M=\frac{\sum fn}{N}$$
where *M* is the average meaning of the frequencies of all populations, and *N* is the number of the populations.

Due to the heterogeneity of population sizes, the average square deviation of the general population was also calculated by Formula (3):(3)$$\sigma =\sqrt{\frac{\Sigma_{i=1}^{N}{\left(fi-M\right)}^{2}}{N}}$$

The statistical significance of the frequencies’ differences was evaluated by criterion t-student by an online service [22,23] and considered as statistically significant under *t* = 1.972 (significance level 0.05).

For the evaluation of the effect of the fixed factors, breed and gender, on the common frequencies of animal carriers in different age populations, we have used the analysis of variance (two-way ANOVA) by using the corresponding software (https://www.statskingdom.com (accessed date 11 March 2021)).

## 3. Results

### 3.1. Development of DNA Tests for CWH and CMD1 Diagnostics

Due to the study the DNA tests for CWH and CMD1, diagnostics have been developed.

For the congenital muscular dystonia diagnostics, we used the PCR-RFLP strategy. The PCR amplicon of 116 bp responding to the wild-type allele. After exposing the restriction endonuclease *DraIII*, the mutant allele was restricted to two DNA fragments with the generation of the visible DNA fragment of 61 bp. Therefore, the animal carriers of the CMDI genetic defects show two DNA fragments of 116 and 61 bp on the electropherogram (Figure 1b). As we can see from Figure 1b, the mutant allele associated with the CMD1 genetic defect was revealed in probe 3, which points out that the animal is the carrier of the CMD1 genetic defect.

The test system for CWH diagnostics based on the AS-PCR method, allowing the simultaneous amplification of wild-type and mutant alleles. The size of the wild-type allele was 141 bp, and the mutant one was 52 bp (Figure 1a).

The Sanger sequencing has shown that the developed test systems are very precious and able to reveal wild-type alleles as mutant alleles associated with the genetic defects that cause the developmental duplication and congenital muscular dystonia of Aberdeen Angus and Belgian Blue cattle (Figure 2).

### 3.2. Genotyping of the Aberdeen Angus Cattle

The Aberdeen Angus populations were genotyped on AM, DD, OS and M1 genetic defects.

We have not found the animal carriers of OS and M1 genetic defects.

However, the AMC and DDC animals were observed in populations 1c, 1d and 2b. The frequency of AMC animals was 1.06–8.49%, and the frequency of DDC animals was 0.71–1.88%, depending on the population (Table 4).

The average frequency of AMC animals among the cow populations was 2.38 ± 0.31%. Among the bull populations, the average frequency was 1.67 ± 0.19%. The average frequency of animal carriers of the DD genetic defect was 0.65 ± 0.07% among the cow groups and 0.90 ± 0.10% among the bull populations.

### 3.3. Genotyping of the Hereford Cattle

The Hereford cattle populations were genotyped on OS, HY, MSUD, ICM and CWH genetic defects. We have not found any animal carriers of OS.

We have found the single animal carrier of ICM and HY genetic defects in population 2 in frequencies of 0.62% for each. The animal carriers of MSUD and CWH genetic defects were found in population 1b with a frequency of 1.08% (Table 5).

### 3.4. Genotyping the Simmental and Belgian Blue Cattle Population

The Simmental cattle populations were genotyped on the OS genetic defect. Any animal carriers of the defect have not been found.

The Belgian Blue cattle were genotyped on M1 and CMD1 genetic defects. All of the animals were carriers of the double-muscling genetic defect (Figure 3), and one of them was a CMD1-carrier that consisted of 0.84%.

### 3.5. Revealing the Trend of the Raising Frequency of Animal Carriers of Genetic Defects in Time

Disposing the DNA samples from cattle populations of different ages from the same farms, we decided to observe the frequency of animal carriers of genetic defects during the years. We compared four populations of Aberdeen Angus cows (1a–d), two populations of Aberdeen Angus bulls (2a,b) and three populations of Hereford heifers (1a–c).

In Figure 4, we presented the results of the evaluation of the increasing of AMC and DDC animal carriers in the Aberdeen Angus cow and bull populations.

We observed increasing frequencies of the genotypes heterozygous on the mutant alleles associated with AM and DD genetic defects in cow and bull of Aberdeen Angus cattle populations. The frequency of AMC animal carriers among the cow population from 2013 to 2015 increased by 1.06%, and from 2015 to 2019, this figure was already 7.43%. For the DD genetic defect, these figures were 0.71% and 1.17%, respectively. In the bull population, the increase of AMC and DDC animal carriers from 2015 to 2019 was 3.33% and 1.80%, respectively.

### 3.6. The Statistical Eveluation of the Data

The statistical analysis of the differences in AMC and DDC animal carrier frequencies showed a significant increase of the DDC animal carriers by 1.17% in the Aberdeen Angus cow population 1d (born 2015) compared with 1c (born 2019) (*t* = 2.21, α = 0.05). The observed differences of AMC animal carriers between 1b (born 2013) and 1c (born 2015) populations (1.06%), 2a (born 2015) and 2b (born 2019) (3.33%), have not shown statistical significance because the meanings of t-criterion were 1.00, 0.89 and 1.5, respectively. The differences in DDC animal carrier between 1b and 1c (0.71%), 2a and 2b (1.80%), were also insignificant (*t* = 1.00).

In addition, we have evaluated the fixed factor influence (breed and gender) on the frequencies of animal carriers in the populations of Aberdeen Angus (AA), Hereford (HE), Simmental (SI) and Belgian Blue (BB) breeds. For this purpose, we counted the quantity of animals free from the mutations. The results have been presented in Table 6.

The null hypothesis (H_0_) proposes no differences in the means of variable A (or B) categories, i.e., the absence of the influence of the breed and gender of animals on the proportion of genetic carriers in the populations. The condition of the H_0_ confirmation was *p*-value > α (0.05). As for Factor A (breed) and B (Gender), the *p*-values consisted of 0.2586 and 0.1327, respectively, which do not reject H_0_ because the chances of a type I error, rejecting a correct H_0,_ is too high (25.86% for factor A and 13.27% for factor B). Therefore, we can conclude the absence of significant influence of breed or gender of animals on the frequencies of genetic defect carriers in populations.

The distribution of cows and bulls of the studied breeds free from the mutations has been demonstrated of Figure 5.

## 4. Discussion

The study of the meat cattle genetic defects since 2017 has shown the existence of animals carrying the mutant alleles linked with the appearance of inherited diseases in their genotypes [24].

The key point in this work, in our opinion, is the development of DNA tests for identifying animals that carry genetic defects in the shortest possible time. The current state of molecular genetics allows for early diagnosis of such animals, and the main task of any genetic laboratory working in the field of beef cattle breeding is to possess fast and accurate diagnostic methods, which must be constantly improved using the latest advances based on the fluorescence detection (for example, real-time PCR).

In this study, we set a goal to comprehensively assess the state of the genetic defects problem in Russia by using already developed and newly created methods for identifying animals that carry hereditary anomalies. Along with analyzing the frequency of occurrence of mutant alleles in beef cattle populations, we also analyzed the dynamics of the spread of certain mutations over time.

The conducted study has shown the frequency of animal carriers of genetic defects in most farms was relatively low. However, an investigation conducted earlier demonstrated the extremely high figure in a single farm: in 2019, we observed cows from a farm in the central region of Russia, and almost 40% of the animals were carrying the mutant allele associated with the DD genetic defect [21]. In addition, we got personal communication about the birth of calves with the signs of the inherited disease from the farmers of Aberdeen Angus and Hereford cattle.

We also observed in one farm in the Northern-West region of Russia trends of increasing animal carriers of genetic defects (AM and DD) in the next generations [21].

The data obtained in the course of this work confirm the relevance of the problem being studied since multiple increases in animals carrying the genetic defect AM were found in two farms, and in some cases, the data was reliable. Relating to the DD genetic defect, we have also observed a non-significant increase in the DDC-carrying animals over the years (Table 3).

At the beginning of the study, we assumed that the Hereford breed was free from harmful mutations associated with congenital pathologies [24]. However, the present investigations have shown the contrasting fact: Among the investigated populations, the single animal carriers of MSUD, ICM, HY and CWH have been revealed (Table 5).

The cows of the Belgian Blue population have the presence of the allele [25] linked with double muscling historically fixed in their genotypes and have shown the presence of the mutant allele associated with CMD1 genetic defect.

The Simmental cattle breed has a double purpose because its feature is the development of milk and meat dependently from the direction of selection and management [26]. This breed is popular in the Russian territory, and so we decided to study the genetic aspect. The animal carriers of the OS genetic defect in the studied population have not been found, and it is insufficient to make any conclusion about the spread of the genetic defects in this breed.

As we have proposed, the fixed factors breed or gender did not influence the quantity of genetic defect carriers in the populations, which was confirmed by a two-way ANOVA.

We are partly reassured by the absence of carrier animals in the Hereford and Simmental bull populations, but the situation may change if there is no control. We would like to emphasize the special importance of genetic testing breeding bulls because due to the widespread use of artificial insemination in cattle breeding, a bull with the seed that dozens of cows are inseminated with can be a significant factor in the spread of hereditary anomalies in the herd.

Therefore, due to the study, we have revealed the animal carriers of the genetic defects and found the tendency to increase in the next generations that point to the danger of the inherited anomalies manifestation and the need to input control measures in the Russian Federation territory.

## 5. Conclusions

It should be noted that the main feature of congenital-inherited diseases is the impossibility of their treatment. Despite the possible development of gene therapy measures, they are very expensive and unbeneficial for the beef cattle breeding. Therefore, currently most animals affected by type I genetic diseases die soon after birth, and those that survive with the manifestation of non-lethal genetic defects of type II require the costs of care and treatment. These are lost profits for the farmers. Additionally, today, the most effective prevention measure is the early identification of the animal carriers of the genetic pathologies and preventing further crossing with animals free from genetic defects. 

Moreover, genetic selection for improved animal health has the potential to complement genetic gains associated with selection for improved productivity traits. Unlike traditional prevention and treatment strategies for diseases, the genetic gain is cumulative and can persist over generations [1]. Therefore, improving the health and immunity of cattle will facilitate an increase in beef production efficiency through a reduction in morbidities and mortalities of animals.

## 6. Patents

The test systems for the AM and DD diagnostics have been patented:Konovalova, E.N., Kostyunina, O.V. and Zinovieva, N.A: Diagnostic technique for the polymorphism of genes AGRN, ISG15 and HES4 causing lethal genetic defect of multiple arthrogryposis of cattle meat breeds. RU Patent 2703396C2. 2019.Konovalova, E.N., Kostyunina, O.V. and Zinovieva, N.A. Method of diagnosing polymorphism of NHLRC2 gene causing genetic defect of duplication of Aberdeen Angus breed cattle development. RU Patent 2715330C2. 2020.

## Figures and Tables

**Figure 1 genes-12-00678-f001:**
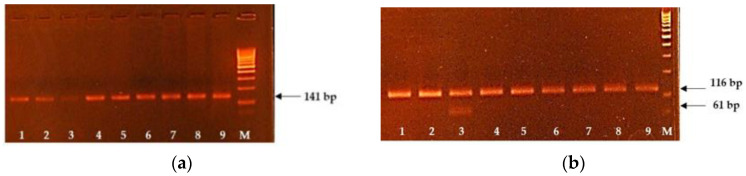
The results of AS-PCR and PCR-RFLP for the diagnostics of CWH and CMD1 genetic defects. (**a**) Lines 1–9 correspond to the animals free from the CWH. (**b**) Lines 1, 2 and 4–9 correspond to the animals free from the CMD1, and line 3 corresponds to the animal carrier of CMD1. M: molecular weight marker (500 bp).

**Figure 2 genes-12-00678-f002:**
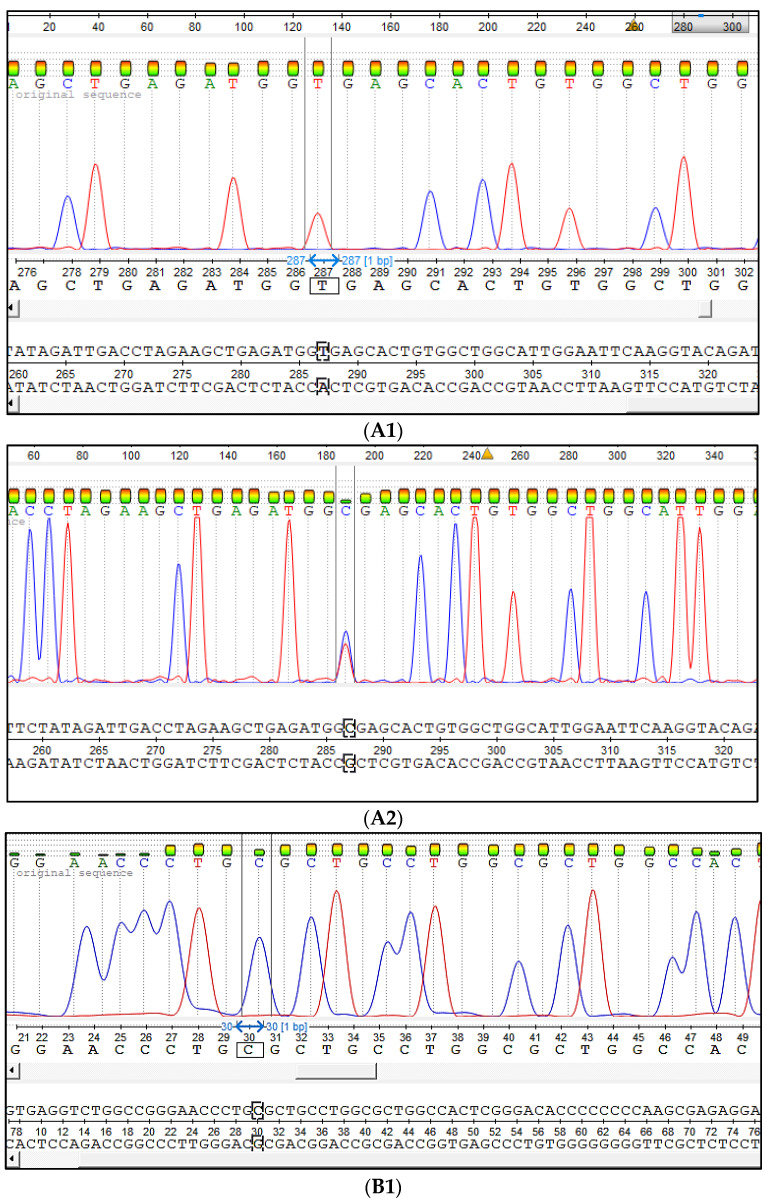
The Sanger sequencing results of DD and CMD1 analysis PCR products. (**A**) The SNP c.932T > C caused DD: (**A1**) The animal homozygous on wild-type allele (DDF), (**A2**) the heterozygous animal carrier of the mutation (DDC). (**B**) The SNP c.1675C > T. SNP caused CMD1: (**B1**) The animal homozygous on wild-type allele (CMD1F), B2-c.932T > C, (**B2**) the heterozygous animal carrier of the mutation (CMD1C).

**Figure 3 genes-12-00678-f003:**
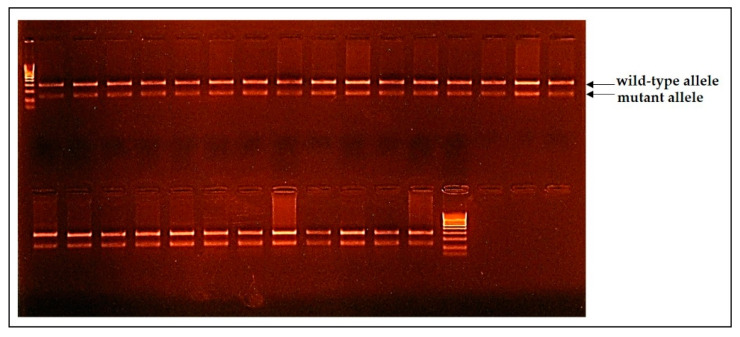
The results of the Belgian Blue cattle genotyping on the M1 genetic defect indicated that all of the animals were carriers of the mutation nt821del *MSTN*.

**Figure 4 genes-12-00678-f004:**
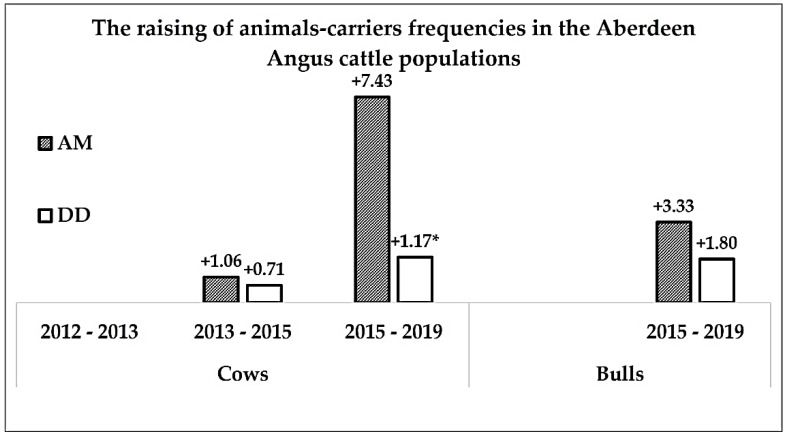
The dynamics of the animal carriers of genetic defects in Aberdeen Angus populations. * statistically significant data.

**Figure 5 genes-12-00678-f005:**
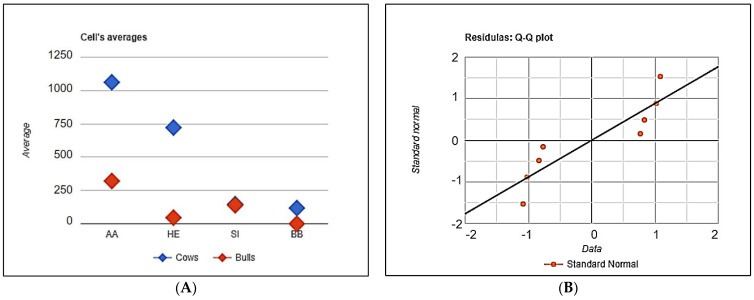
The distribution of the animals free of genetic defects in Aberdeen Angus (AA), Hereford (HE), Simmental (SI) and Belgian Blue (BB) cattle, depending on the gender and breed. (**A**) The distribution of the defects free cows and bulls, (**B**) The distribution of the standard deviations.

**Table 1 genes-12-00678-t001:** The short characteristics of some gene mutations and caused genetic defects of meat cattle breeds.

Congenital Disease	Lethal Type	Clinical Signs	Caused Mutation	Breed
Arthrogriposis multiplex (AM)	I	Affected calves are small due limited muscle development, have bent and twisted spine, rigid and hyperextended rear legs leading to calving difficulties. The calves are stillborn or die in a few hours after birth [5].	The 23363-bp deletion of the entirety *ISG15*, one or both of the 5’ regulatory region of *HES4* and the first two exons of *AGRN* gene [6].	Aberdeen Angus
Osteopetrosis (OS)	I	Affected calves were born 3 weeks premature and had characteristics of skeletal defects. Lesions in the head included inferior brachygnathia, impacted premolars, incomplete and misaligned incisors, a protuberant tongue, fixed temporomandibular joints and a flat calvarium [7].	The 2781–bp deletion mutation in *SLC4A2* [8].	Aberdeen Angus, Friesian, Hereford, Simmental
Developmental duplication (DD)	II	The signs of polymelia and many another DD phenotypes due neural tube defect [9].	SNP c.932T > C within exon 5 of *NHLRC2* gene [10].	Aberdeen Angus
Double muscling (M1)	II	The increase of muscle mass by approx. 20%, resulting in substantially higher meat yield, a higher proportion of expensive cuts of meat, and lean and very tender meat, for which a substantial premium is paid. Disadvantages: greatly increased incidence of calving difficulties [11].	The 11-bp deletion (of nucleotides 821 to 831) *MSTN* gene [12].	Aberdeen Angus, Belgian Blue
Hypotrichosis (HY)	II	Affected cattle have partial absence of hair at birth over all or parts of the body: often on the poll, brisket, neck and legs. The hair can be very short, fine or kinky, which may fall out leaving bare spots, and the tail switch can be underdeveloped. Affected animals are more vulnerable to environmental stress, skin infections, pests, sunburn, cold stress and have a decreased economic value [9].	A small deletion (c.334delTGTGCCCA) in *KRT71* gene [13].	Hereford
Maple syrup urine disease (MSUD)	I	Some affected calves are stillborn. Those born alive look normal but exhibit neurological symptoms within 24 h. Their condition will rapidly deteriorate with ataxia, an inability to walk and death within 96 h after birth. The most telling symptom is that the animals have sweet-smelling urine [9].	SNP 248C > T in the leader region of the *BCKDHA* gene [14].	Hereford
Inherited congenital myoclonus (ICM)	I	Affected animals often appear normal but have spontaneous muscle spasms and whole body rigidity in response to stimulation. When laying down, the back legs are often crossed. When assisted to a standing position, the handlers touch can cause full body rigidity and a sawhorse position [15].	SNP c.156C > A in *GLRA1* gene [16].	Hereford
Cardiomyopathy with wool hair (CWH)	I	Neonatal ocular keratitis develops in some cases, and death usually occurs within the first 12 weeks of life. Affected calves can be identified at birth by their distinctive woolly haircoat and by signs of rapidly progressing cardiac dysfunction, including arrhythmias. Neonatal ocularkeratitis develops in some cases. The animal usually dies within the first 12 weeks of life [17].	The 7-bp duplication (c.956_962dup7) in exon 6 13-exon gene *PPP1R13L* [18].	Hereford
Congenital muscular dystonia type 1 (CMD1)	I	Animals have muscle myotonia, which results in an inability to flex limbs and injurious falling. They also experience fatigue upon stimulation. The mutation causes a disorder in muscle function due to a defect in the Ca^2^ pump [19].	SNP c.1675C > T *ATP2A1* gene [20].	Belgian Blue, Dutch Improved Red and White

**Table 2 genes-12-00678-t002:** The description of the research material.

Breed	Population	Technological Group	Year of Birth	*n*
Aberdeen Angus	1a	Cows	2012	85
	1b	Cows	2013	67
	1c	Cows	2017	281
	1d	Heifers	2019	106
	2a	Bulls	2015	72
	2b	Bulls	2019	90
Hereford	1a	Heifers	2016	61
	1b	Heifers	2017	93
	1c	Heifers	2020	46
	2	Cows	2015	162
	3	Bulls	2020	23
Simmental	1	Heifers	2018	147
	2	Bulls	2018	41
	3	Bulls	2019	59
	4	Bulls	2020	39
Belgian Blue	1	Cows	2017	118

**Table 3 genes-12-00678-t003:** Primer sequences for genetic defects diagnostics.

Genetic Defect	Name of the Primer	5′-3′ Sequence	Annealing Temperature, °C
AM	AMV1	cgaaagccttctttccactg	66.0
	AMV2	ttctgcaggcaagaacactg	
	AMV3	gaatgccacttcctcctctg	
OS	OSF	agcccctacagtcacagtca	60.0
	OSRn	agcagcagagatcagcttgg	
	OSRm	ccgaccccctcacattcaaa	
DD	DD1	agaggcatgatgaaggcgag	61.0
	DD3	ccaaggggaactaatgggct	
M1	MSTN821F	tgaggtaggagagtgttttggg	60.0
	MSTN821Rn	cctctggggtttgcttggt	
	MSTN821Rm	acagcatcgagattctgtcaca	
HY	HY_F	cggaagtcggagcctttaca	65.0
	HY_Rn	acgcactttctggatctcgg	
	HY_Rm	ccaggtcagttgggcacat	
CWH	CWH_F	actctgccccgcaattacaa	63.5
	CWH_Rn	catggggatgcgactgacag	
	CWH_Rm	cctgtcgcctgtcgttgg	
CMD1	CMD_F	ggccggtgaaggagaagatt	58.0
	CMD_R	gaccatctcctctcgcttgg	

**Table 4 genes-12-00678-t004:** The results of the Aberdeen Angus cattle genotyping on Arthrogriposis multiplex (AM) and Developmental duplication (DD) genetic defects.

Population		Frequencies of Animal Carriers, %		
	AM	M ± σ	DD	M ± σ
1a	0.00	2.38 ± 0.31	0.00	0.65 ± 0.07
1b	0.00		0.00	
1c	1.06		0.71	
1d	8.49		1.88	
2a	0.00	1.67 ± 0.19	0.00	0.90 ± 0.10
2b	3.33		1.80	

**Table 5 genes-12-00678-t005:** The results of the Hereford cattle genotyping on Osteopetrosis (OS), Maple syrup urine disease (MSUD), inherited congenital myoclonus (ICM), Hypotrichosis (HY) and Cardiomyopathy with wool hair (CWH) genetic defects.

Population	Frequencies of Animal Carriers, %							
	MSUD	M ± σ	ICM	M ± σ	HY	M ± σ	CWH	M ± σ
1a	0.00	0.27 ± 0.05	0.00	0.16 ± 0.03	0.00	0.16 ± 0.03	0.00	0.27 ± 0.05
1b	1.08		0.00		0.00		1.08	
1c	0.00		0.00		0.00		0.00	
2	1.88		0.62		0.62		0.00	
3	0.00		0.00		0.00		0.00	

**Table 6 genes-12-00678-t006:** The results of the two-way ANOVA of the obtained data.

Source	DF *	Sum of Square (SS)	Mean Square (MS)	F Statistic (df_1_,df_2_)	*p*-Value
Factor A—Breed	3	483,113.375	161,037.7917	2.2744 (3,3)	0.2586
Factor B—Gender	1	297,606.125	297,606.125	4.2032 (1,3)	0.1327
Error	3	212,416.375	70,805.4583		
Total	7	993,135.875	141,876.5536		

DF *—freedom degree.

## Data Availability

The original data of the paper are available upon request from the corresponding author.

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
