# Peer review of "The Molecular Bases Study of the Inherited Diseases for the Health Maintenance of the Beef Cattle"

_genes, 2021, doi:10.3390/genes12050678_

Round 1

Reviewer 1 Report

The use of PCR-RFLP should not be advised since rare alleles may influence the outcome. For that the authors should have included a control population for which sanger sequencing of the target region would assure that these animals are not carriers of the diseases mutations, without a control population you cannot assure the accuracy of the test. Then after the control a random assay should be developed to assure the replication of rresults to any other samples.

Author Response

Dear Reviewer,

Thank you for the attention to our manuscript. We agreed with it, carried out the additional work to eliminate this defect, and made appropriate corrections to the manuscript.

Reviewer 2 Report

The recognition of the genetic traits (within the ICAR/INTERBULL/INTERBEEF terminology) could be a fast and efficient way to recognize inherited diseased and positively selected traits, i.e. milk composition (beta and kappa caseins or DGAT1 genes). The portion of the information flowing from the casual mutation could have positive incoming locally to have the healthy animals and selling the high-quality product, and widely to do not spread the word genetic defects by the selling the genetic material. The study aimed to develop DNA-tests for genetic traits diagnostics and determine the animal carriers and their frequencies tracking over time.

The introduction part was mainly focused on the description of the selected traits. Most of the knowledge was taken from the omia.angis.org and ncbi.nih.gov.nlm traits/ diseases databases. There is no word about the technics used in the experiment and, in general, about technics used widely over the world. This issue is significant, especially in the genomic selection era. In many cases, the recognition of the genetic traits (based on casual mutations) could be an addition to the genomic breeding value estimation. Moreover, equivalent tools are prepared and dedicated to this porpoise, like SNP microarrays, which evolved to multi porpoise breeding tools. The best example could be the Illumina EuroGenomic EuroGMD chip witch saved to GEBV estimation, parentage verification, imputation and finally to genetic defects identification. There is no description of the lethality of the genetic traits but only a mark in Table 1.

The material and methods section describes the used animals (1351 samples) and molecular biology technics (AS-PCR and PCR-RFLP). However, statistical analysis looks highly trivial. Instead of the t-test, the chi2-test should be used or two-way ANOVA (linear model) considering fixed effects like sex, subpopulation and birth across Hereford and Angus population. The results and discussion sections were described well based on the proposed M&M. However, the figure's quality must be better. I also have a huge question mark based on the thesis that PCR technics were able to "identifying animals that carry genetic defects in the shortest possible time." Is it the thru assumption based on the broad-scale SNP microarray genotyping/HRM RT-PCR opposite the PCRs. Maybe the direction of the used technics should goest to the HRM SNP genotyping analysis.

The general question is more based on the beef cattle breeding system across Russia. The results showing only one cattle station and the therm of the genetic defect monitoring carriers are not efficient. There are many breeding marks, like how the new genetic material is incorporated on the farm, how it looks like animals mating, and how it looks like the pedigree structure of used animals. Some of the general questions find the semi answers in Discussions. However, there were not supplied by the experimental study. Moreover, the English correction is warranted.

Author Response

Dear reviewer,

thank you very much for the attention and detailed analysis of our manuscript. We have tried to answer your questions exhaustively and revised the manuscript according to your comments.

Round 2

Reviewer 2 Report

I recommend the paper to ACCEPTATION in the current form and finalize for the post review processes.